# Time-Reversal Symmetric ODE Network

**In Huh**[1,2], **Eunho Yang**[3,4], **Sung Ju Hwang**[3,4], **Jinwoo Shin**[3,5]
[1]Samsung Advanced Institute of Technology
[2]CSE Team, DIT Center, Samsung Electronics
[3]Graduate School of AI, Korea Advanced Institute of Science and Technology (KAIST)
[4]School of Computing, KAIST
[5]School of Electrical Engineering, KAIST
in.huh@samsung.com, yangeh@gmail.com, {sjhwang82, jinwoos}@kaist.ac.kr

## Abstract

*Time-reversal symmetry*, which requires that the dynamics of a system should not change with the reversal of time axis, is a fundamental property that frequently holds in classical and quantum mechanics. In this paper, we propose a novel loss function that measures how well our ordinary differential equation (ODE) networks comply with this time-reversal symmetry; it is formally defined by the discrepancy in the time evolutions of ODE networks between forward and backward dynamics. Then, we design a new framework, which we name as *Time-Reversal Symmetric ODE Networks (TRS-ODENs)*, that can learn the dynamics of physical systems more sample-efficiently by learning with the proposed loss function. We evaluate TRS-ODENs on several classical dynamics, and find they can learn the desired time evolution from observed noisy and complex trajectories. We also show that, even for systems that do not possess the full time-reversal symmetry, TRS-ODENs can achieve better predictive performances over baselines.

## 1 Introduction

Recent advances in artificial intelligence allow researchers to recover laws of physics and predict dynamics of physical systems from observed data by utilizing machine learning techniques, e.g., evolutionary algorithms [37, 29], sparse optimizations [35, 4], Gaussian process regressions [40, 8], and neural networks [18, 1, 15, 46, 34]. Among various models, the neural networks are considered as one of the most powerful tools to model complicated physical phenomena, owing to their remarkable ability to approximate arbitrary functions [17]. One notable aspect of the observations in physical systems is that they manifest some fundamental properties including conservation or invariance [14, 2]. However, it is not straightforward for neural networks to learn and model the embedded physical properties from observed data only. Consequently, they often overfit to short-term training trajectories and fail to predict the long-term behaviors of complex dynamical systems [15, 46].

To overcome these issues, it is important to introduce appropriate inductive biases based on knowledge of physics, dynamics and their properties [46, 34]. Common approaches to incorporate physics-based inductive bias include modifying neural network architectures or introducing regularization terms based on specialized knowledge of physics and natural sciences [31, 28, 38, 39]. These methods demonstrate impressive performance on their target problems, but such a problem-specific model suffers from generalizing across domains. Namely, they can be used only when the governing physics for the target domain is exactly known, e.g., Navier-Stokes equation for fluid mechanics [31, 28]. As for more general approaches, the authors in [6, 5] propose the *ordinary differential equation (ODE) networks*, which view the neural networks as parameterized ODE functions. They are shown to be able to represent the vast majority of dynamical systems with higher precision over vanilla recurrent neural networks and their variants [6, 5], but are still unable to learn underlying physics such as the

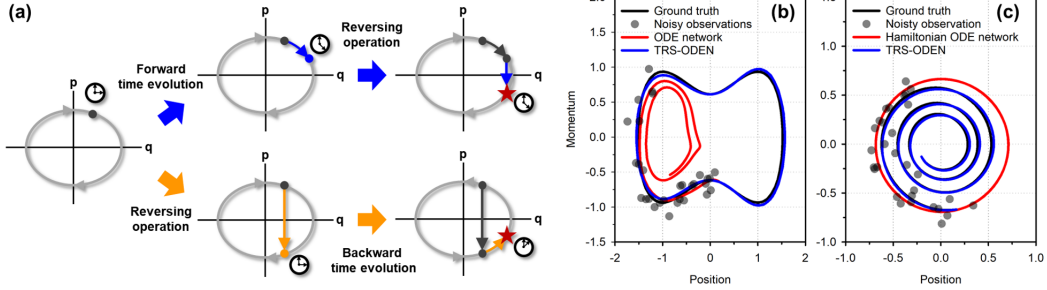

Figure 1: (a) **Time-reversal symmetry of dynamical systems.** The gray ellipse is a phase space trajectory, which does not change under $t \mapsto -t$. The reversing of forward time evolution (blue arrows) of an arbitrary state should yield an equal state to what is estimated by the backward time evolution of the reversed state (orange arrows). For more mathematical details, see Section 3.2. Examples of (b) non-linear and (c) non-ideal dynamical systems modeled by various ODE networks including TRS-ODENs. TRS-ODENs can learn appropriate long-term dynamics from noisy and short-term training samples.

law of conservation [15]. Recent works [15, 46, 34, 7, 43] apply the Hamiltonian mechanics to ODE networks, and succeed in enforcing the energy conservation as well as the accurate time evolution of classical conservative systems. However, these Hamiltonian ODE networks have inherent limitations that they cannot be applied to non-conservative systems, since the Hamiltonian structures require to strictly conserve the total energy [15].

To address such limitations of existing works on modeling classical dynamics, we introduce a physics-inspired, general, and flexible inductive bias, *symmetries*. It is at the heart of the physics: the laws of physics are invariant under certain transformations in space and time coordinates, thus show the universality [12, 27]. For example, the classical dynamics possess the *time-reversal symmetry*, which means the classical equations of motion should not change under the transformation of time reversal: $t \mapsto -t$ [23, 32, 42] (see Figure 1). Therefore, if the target underlying physics being approximated has some symmetries, it is natural that the approximated physics using neural networks should also comply with these properties. Motivated by this, we feed the symmetry as an additional information to help neural networks learn the physical systems more efficiently.

Specifically, we focus on the *time-reversal symmetry* of classical dynamics described above, due to its simplicity and popularity. We propose a new ODE learning framework, which we refer to as *Time-Reversal Symmetry ODE Network (TRS-ODEN)*[1], that utilize the time-reversal symmetry as a regularizer in training ODE networks, by unifying recent studies of ODE networks [7] and classical symmetry theory for ODE systems [23]. Our scheme can be easily implemented with a small modification of codes for conventional ODE networks, and is also compatible with extensions of ODE networks, such as Hamiltonian ODE networks [46, 34, 7]. It can be used to predict many branches of physical systems, because the isolated classical and quantum dynamics exhibit the perfect time-reversal symmetry [23, 33]. Moreover, even for the case when the full time-reversal symmetry are broken [23], e.g., in the presence of the entropy production [30] through heat or mass transfer, we also show that TRS-ODENs are beneficial to learn such system by annealing the strength of the proposed regularizer appropriately. This flexibility with regard to the target problem is the main advantage of the proposed framework, in contrast to prior methods, e.g., only for suitable explicitly conservative systems [15].We validate our proposed model in several domains including synthetic Duffing oscillators [22] (see Section 4.1), real-world coupled oscillators [37] (see Section 4.2), and reversible strange attractors [41] (see Section 4.3). In summary, our contribution is threefold:

- We propose a novel loss function that measures the discrepancy in the time evolution of ODE networks between forward and backward dynamics, thus estimates whether the ODE networks are time-reversal symmetric or not.

- We show ODE networks with the proposed loss, coined TRS-ODENs, achieve better predictive performance than baselines, e.g., from 50.81 to 10.85 for non-linear oscillators.

- We validate even for time-irreversible systems, the proposed framework still works well compared to baselines, e.g., from 3.68 to 0.12 in terms of error for damped oscillators.

## 2 Background and Setup

### 2.1 Predicting dynamical systems

In a dynamical system, its states evolve over time according to the governing time-dependent differential equations. The state is a vector in the phase space, which consists of all possible positions and momenta of all particles in the system. If one knows the governing differential equation and initial state of the system, the future state is predictable by solving the equation analytically or numerically. On the other hand, if one does not know the exact governing equation, but has some state trajectories of the system, one can try to model the dynamical system, e.g., by using neural networks. More specifically, one can build a neural network whose input is current state (or trajectory) and the output is the next state, from the perspective of the sequence prediction. However, such a method may overfit to short-term training trajectories and fail to predict the long-term behaviors [46]. It is also not straightforward to predict the continuous-time dynamics, because neural network models typically assume the discrete time-step between states [15]. Neural ODE and its applications [6, 5, 15, 46, 34, 7, 43], alias ODE networks (ODENs), tackle these issues by learning the governing equations, rather than the state transitions directly. Moreover, some of them use special ODE functions such as Hamilton's equations to incorporate physical properties to neural network structurally [15, 46, 34, 7, 43]. In the rest of this section, we briefly review ODENs and Hamiltonian ODE networks (HODENs), which are closely related to our work.

### 2.2 ODE networks (ODENs) for learning and predicting dynamics

We consider dynamics of state $\mathbf{x}$ in phase space $\Omega$ ($= \mathbb{R}^{2n}$, in classical dynamics[2]) given by:

$$\frac{d\mathbf{x}}{dt} = f(\mathbf{x}) \quad \text{for } t \in \mathbb{R}, \ \mathbf{x} \in \Omega, \ f : \Omega \mapsto T\Omega. \tag{1}$$

The continuous time evolution between arbitrary two time points $t_i$ and $t_{i+1}$ by (1) is equal to:

$$\mathbf{x}(t_{i+1}) = \mathbf{x}(t_i) + \int_{t_i}^{t_{i+1}} f(\mathbf{x})dt. \tag{2}$$

The recent works [46, 34, 6, 7] propose the ODENs, which represent the ODE functions $f$ in (1) by neural networks and learn the unknown dynamics from data. For ODENs, fully-differentiable numerical ODE solvers are required to train the black-box ODE functions, e.g., Runge-Kutta (RK) method [11] or symplectic integrators such as leapfrog method [24]. With an ODE solver, say `Solve`, one can estimate the time evolution by ODENs:

$$\tilde{\mathbf{x}}(t_{i+1}) = \texttt{Solve}\{\tilde{\mathbf{x}}(t_i), f_\theta, \Delta t_i\}, \ \tilde{\mathbf{x}}(t_0) = \mathbf{x}(t_0), \tag{3}$$

where $f_\theta$ is a $\theta$-parameterized neural network, $\tilde{\mathbf{x}}(t_i)$ is a prediction of $\mathbf{x}(t_i)$ using ODENs, $\Delta t_i = t_{i+1} - t_i$ is a time-step, and $\mathbf{x}(t_0)$ is a given initial value. Given observed trajectory $\mathbf{x}(t_1), ..., \mathbf{x}(t_T)$, ODENs can learn the dynamics by minimizing the loss function $\mathcal{L}_{\text{ODE}} \equiv \sum_{i=0}^{T-1} \|\texttt{Solve}\{\tilde{\mathbf{x}}(t_i), f_\theta, \Delta t_i\} - \mathbf{x}(t_{i+1})\|_2^2$. We omit the sample-wise mean for the simple notation.

### 2.3 Hamiltonian ODE networks (HODENs)

The Hamiltonian mechanics describes the phase space equations of motion for conservative systems by following two first-order ODEs called Hamilton's equations [14]:

$$\frac{d\mathbf{q}}{dt} = \frac{\partial \mathcal{H}(\mathbf{q}, \mathbf{p})}{\partial \mathbf{p}}, \ \frac{d\mathbf{p}}{dt} = -\frac{\partial \mathcal{H}(\mathbf{q}, \mathbf{p})}{\partial \mathbf{q}}, \tag{4}$$

where $\mathbf{q} \in \mathbb{R}^n$, $\mathbf{p} \in \mathbb{R}^n$, and $\mathcal{H} : \mathbb{R}^{2n} \mapsto \mathbb{R}$ are positions, momenta, and Hamiltonian of the system, respectively. Recent works [46, 34, 7] apply the Hamilton's equations to ODENs, by parameterizing the Hamiltonian as $\mathcal{H}_\theta$, and replacing $f_\theta(\mathbf{q}, \mathbf{p})$ to the gradients of $\mathcal{H}_\theta$ with respect to inputs $(\mathbf{p}, \mathbf{q})$ according to (4). Thus, the time evolution of HODENs is equal to:

$$(\tilde{\mathbf{q}}(t_{i+1}), \tilde{\mathbf{p}}(t_{i+1})) = \texttt{Solve}\{(\tilde{\mathbf{q}}(t_i), \tilde{\mathbf{p}}(t_i)), (\partial \mathcal{H}_\theta/\partial \mathbf{p}, -\partial \mathcal{H}_\theta/\partial \mathbf{q}), \Delta t_i\}. \tag{5}$$

HODENs show better predictive performance for conservation systems. Furthermore, they can learn the underlying law of conservation of energy automatically, since they fully exploit the nature of the Hamiltonian mechanics [15]. However, a fundamental limitation of HODENs is that they do not work properly for the non-conservative systems [15], because they always conserve the energy.

# 3 Time-Reversal Symmetry Inductive Bias for ODENs

## 3.1 Target systems

Before introducing the time-reversal symmetry, we briefly explain two perspectives of the classical dynamical systems: *conservative* and *reversible*. The former is the system that its Hamiltonian does not depend on time explicitly, i.e., $\partial \mathcal{H} / \partial t = 0$. The latter is the system that possesses the time-reversal symmetry, whose mathematical details will be discussed in the following section.

**Conservative and reversible systems.** All conservative systems that their Hamiltonians satisfy $\mathcal{H}(\mathbf{q}, \mathbf{p}) = \mathcal{H}(\mathbf{q}, -\mathbf{p})$ are also reversible [23]. It means that many kinds of classical dynamics are both conservative and reversible[3]. For these systems, both Hamiltonian and time-reversal symmetry inductive biases are appropriate. Furthermore, combining two inductive biases can improve the sample efficiency of a learning scheme.

**Non-conservative and reversible systems.** It is noteworthy that reversible systems are not necessarily conservative systems. Some examples about non-conservative but reversible systems can be found in [23, 32]. Clearly, baselines such as HODENs that enforce conservative property would break down in this environment. On the other hand, our scheme, named TRS-ODEN, presented in Section 3.3 would accurately model the dynamics of given data by exploiting time-reversal symmetry.

**Non-conservative and irreversible systems.** Under interactions with environments, the dynamical systems become non-conservative and often irreversible[4]. Depending on the intensity of such interactions, the Hamiltonian or time-reversal symmetry inductive bias can be beneficial or harmful. HODENs strictly enforce the conservation, thus they are not suitable for this [15]. On the other hand, TRS-ODENs are more flexible, since they use the inductive bias as a form of regularizer, which is easily controlled via hyper-parameter tuning [36].

## 3.2 Time-reversal symmetry in dynamics

First-order ODE systems (1) are said to be *time-reversal symmetric* if there is an invertible transformation $R : \Omega \mapsto \Omega$, that reverses the direction of time:

$$\frac{dR(\mathbf{x})}{dt} = -f(R(\mathbf{x})), \tag{6}$$

where $R$ is called *reversing operator* [23]. Comparing (1) and (6), one can find that the equation is invariant under the transformations of phase space $R$ and time-reversal $t \mapsto -t$. For notational simplicity, let's introduce a time evolution operator $U_\tau : \Omega \mapsto \Omega$ for (1) as follows [23]:

$$U_\tau : \mathbf{x}(t) \mapsto U_\tau(\mathbf{x}(t)) = \mathbf{x}(t + \tau), \tag{7}$$

for arbitrary $t, \tau \in \mathbb{R}$. Then, in terms of the time evolution operator (7), (6) imply:

$$R \circ U_\tau = U_{-\tau} \circ R, \tag{8}$$

which means that *the reversing of the forward time evolution of an arbitrary state should be equal to the backward time evolution of the reversed state* (see Figure 1).

In classical dynamics, generally, even-order and odd-order derivatives with respect to $t$ are respectively preserved and reversed under $R$ [23, 32]. For example, consider a conservative and reversible Hamiltonian $\mathcal{H}(\mathbf{q}, \mathbf{p}) = \mathcal{H}(\mathbf{q}, -\mathbf{p})$, as mentioned in Section 3.1. Because $\mathbf{q}$ and $\mathbf{p}$ are respectively zeroth and first order derivatives with respect to $t$, $R$ is simply given by $R(\mathbf{q}, \mathbf{p}) = (\mathbf{q}, -\mathbf{p})$. In this case, one can easily check the Hamilton's equations (4) are invariant under $R$ and $t \mapsto -t$. We use this classical definition $R(\mathbf{q}, \mathbf{p}) = (\mathbf{q}, -\mathbf{p})$ in the remainder of the paper, unless otherwise specified.

## 3.3 Time-reversal symmetry ODE networks (TRS-ODENs)

Inspired from ODENs (3) and time-reversal symmetry (8), here we propose a novel *time-reversal symmetry loss*. First, the backward time evolution of the reversed state for ODENs is equal to:

$$\tilde{\mathbf{x}}_R(t_{i+1}) = \texttt{Solve}\{\tilde{\mathbf{x}}_R(t_i), f_\theta, -\Delta t_i\}, \, \tilde{\mathbf{x}}_R(t_0) = R(\tilde{\mathbf{x}}(t_0)). \tag{9}$$

Then, using (3) and (9), we define the time-reversal symmetry loss $\mathcal{L}_{\text{TRS}}$ as an ODEN version of (8):

$$\mathcal{L}_{\text{TRS}} \equiv \sum_{i=0}^{T-1} \| R(\texttt{Solve}\{\tilde{\mathbf{x}}(t_i), f_\theta, \Delta t_i\}) - \texttt{Solve}\{\tilde{\mathbf{x}}_R(t_i), f_\theta, -\Delta t_i\}\|_2^2. \tag{10}$$

Note that we assume the system is autonomous. For non-autonomous systems, see Section A in the supplementary material. Finally, we define the TRS-ODEN as a class of ODENs whose loss function $\mathcal{L}_{\text{TRS-ODEN}}$ is given by the sum of the ODE error $\mathcal{L}_{\text{ODE}}$ and symmetry regularizer $\mathcal{L}_{\text{TRS}}$ as follows[5]:

$$\mathcal{L}_{\text{TRS-ODEN}}(\mathbf{x}(t), \tilde{\mathbf{x}}(t), \tilde{\mathbf{x}}_R(t), R, \theta) \equiv \mathcal{L}_{\text{ODE}}(\mathbf{x}(t), \tilde{\mathbf{x}}(t), \theta) + \lambda \cdot \mathcal{L}_{\text{TRS}}(\tilde{\mathbf{x}}(t), \tilde{\mathbf{x}}_R(t), R, \theta), \tag{11}$$

where $\lambda \geq 0$ is a hyper-parameter. It is noteworthy that $\lambda$ can be also a function of time $t$, especially when dealing with irreversible systems. It is owing to the heuristic that the irreversible term is also a function of $(t, \mathbf{x}(t))$, i.e., although target dynamics do not possess the full time-reversal symmetry, they can be partially reversible when the irreversible term becomes negligible at certain time points.

## 4  Experiments

**Default model setting.** We compare three models: vanilla ODEN, HODEN, and TRS-ODEN. A single neural network $f_\theta(\mathbf{q}, \mathbf{p})$ is used for ODENs and TRS-ODENs, while HODENs consist of two neural networks $K_{\theta_1}(\mathbf{p})$ and $V_{\theta_2}(\mathbf{q})$, i.e., separable $\mathcal{H}_\theta(\mathbf{q}, \mathbf{p}) = K_{\theta_1}(\mathbf{p}) + V_{\theta_1}(\mathbf{q})$ [7]. Also, we use the leapfrog integrator for Solve [7], unless otherwise specified. The maximum allowed trajectory length at training phase is set to 10. If training trajectories are longer that 10, we divide them properly. We train models by using the Adam [19] with initial learning rate of $2 \times 10^{-4}$ during 5,000 epochs. We use full-batch training because training sample sizes are quite small, except for Experiment VI.

**Performance metric.** As primary performance metrics, we use the mean-squared error (MSE) between test ground truths and models' predictive phase space trajectories as well as total energies[6]. The predictive trajectories are obtained by recursively solving (3) or (5), thus errors accumulate and diverge over time if the models do not learn the accurate time evolution.

### 4.1  Experiment I-IV: Learning the Duffing oscillators

Firstly, we focus on the Duffing oscillator [22], a generalized model of oscillators, that given by[7]:

$$\frac{d\mathbf{q}}{dt} = \mathbf{p}, \; \frac{d\mathbf{p}}{dt} = -\alpha\mathbf{q} - \beta\mathbf{q}^3 - \gamma\mathbf{p} + \delta\cos(t), \tag{12}$$

where $\alpha$, $\beta$, $\gamma$, and $\delta$ are scalar parameters that determine the linear stiffness, non-linear stiffness, damping, and driving force terms, respectively. For non-zero parameters, Duffing oscillators are neither conservative nor reversible. Furthermore, they often exhibit chaotic behaviors [22]. However, the characteristics of Duffing oscillator can be changed greatly by adjusting parameters. Thus, by using these coupled equations, we can simulate several dynamical systems mentioned in Section 3.1.

Unless otherwise stated, we generate 50 trajectories each for training and test sets. For each trajectory, The initial state $(\mathbf{q}(t_0), \mathbf{p}(t_0))$ is uniformly sampled from annulus in $[0.2, 1]$. The lengths of training and test trajectories are 30 and 200, respectively, while the time-step is fixed at 0.1, i.e., $\Delta t_i = 0.1$ for all $i$. Thus, we can evaluate whether the models can mimic long-term dynamics. We add Gaussian noise $0.1n, n \sim \mathcal{N}(0, 1)$ to training set. We use the fourth order RK method to get trajectories.

In Experiment I and II, we consider conservative and reversible systems, where we show that TRS-ODENs are comparable with or even outperform HODENs. Moreover, we confirm combining HODENs and time-reversal symmetry loss can lead further improvement for these systems. Then, we evaluate proposed framework for a non-conservative and reversible system in Experiment III. Finally, in Experiment IV, we validate our proposed framework for a non-conservative and irreversible damped system. HODENs cannot learn this system because of their strong tendency to conserve the energy, as previously reported in [15]. We demonstrate TRS-ODENs can learn this system flexibly.

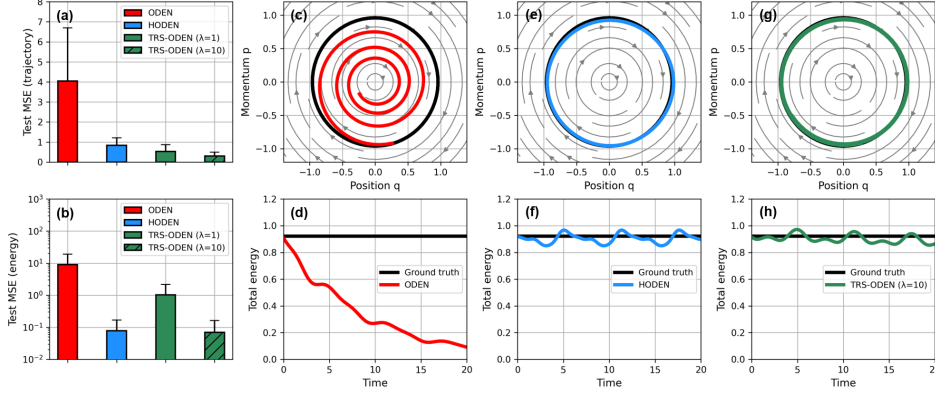

Figure 2: **Summary of Experiment I.** (a-b) Test (a) trajectory MSE and (b) energy MSE across the models. (c-h) Sampled test trajectory and its total energy for the (c-d) ODEN, (e-f) HODEN, and (g-h) TRS-ODEN.

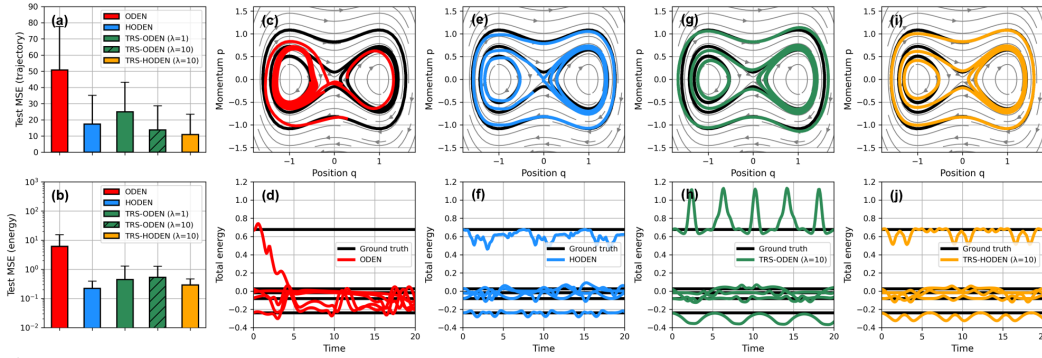

Figure 3: **Summary of Experiment II.** (a-b) Test (a) trajectory MSE (b) and energy MSE across the models. (c-h) Sampled test trajectories and their total energies for the (c-d) ODEN, (e-f) HODEN, (g-h) TRS-ODEN, and (i-j) TRS-HODEN.

**Experiment I: Simple oscillator.** For a toy example, we choose a simple oscillator, i.e., $\alpha = 1$, $\beta = \gamma = \delta = 0$. We use single hidden layer neural networks consists of 1,000 hidden units and `tanh` activations for all models. Figure 2 (a-b) show that the TRS-ODEN with $\lambda = 10$[8] outperforms both ODEN and HODEN. For qualitative analysis, we plot a test trajectory and its total energy (see Figure 2 (c-h)). It shows the TRS-ODEN can learn the energy conservation as well as accurate dynamics.

**Experiment II: Non-linear oscillator.** As a more interesting problem, we choose the undamped and unforced non-linear oscillator, i.e., $\alpha = -1$, $\beta = 1$, $\gamma = \delta = 0$. We use neural networks consist of two hidden layers with 100 units and `tanh` activations.

In this experiment, the TRS-ODEN outperforms HODEN in terms of the trajectory MSE, and vice-versa for total energy MSE (see Figure 3 (a-b)). For qualitative analysis, we sample five trajectories and their energy values (see Figure 3 (c-h)). It shows the HODEN fails to learn time evolution especially near the origin point, while the TRS-ODEN shows undesirable peaks in energy. This room for improvement leads us to combining the HODEN and TRS-ODEN, the *Time-Reversal Symmetric Hamiltonian ODE Network (TRS-HODEN)*[9]. After estimation, We

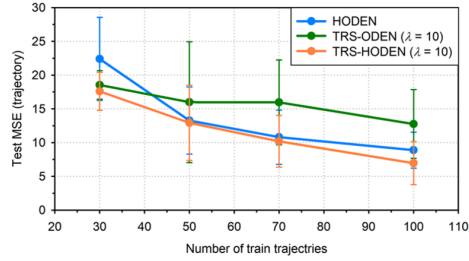

Figure 4: Test MSE *vs.* the number of training samples across models. Means and error bars are calculated by repeating the experiment 5 times, with varying datasets.

find that the TRS-HODEN can achieve almost same performance as HODEN in terms of energy MSE, and clearly outperform baselines for trajectory MSE (see Figure 3 (a-b) and (i-j)). Furthermore,

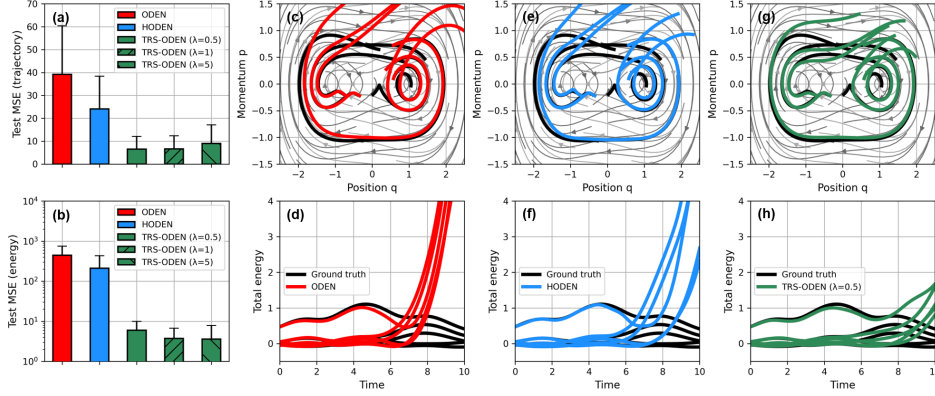

Figure 5: **Summary of Experiment III.** (a-b) Test (a) trajectory MSE (b) and energy MSE across the models. (c-h) Sampled test trajectories and their total energies for the (c-d) ODEN, (e-f) HODEN, and (g-h) TRS-ODEN.

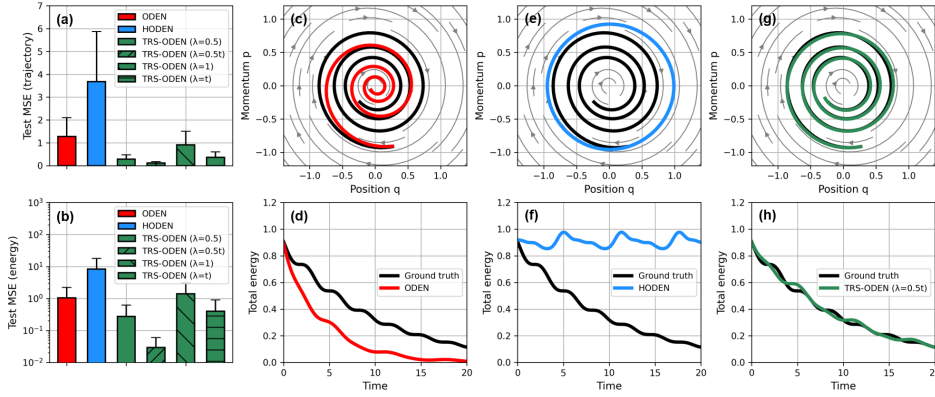

Figure 6: **Summary of Experiment IV.** (a-b) Test (a) trajectory MSE and (b) energy MSE across the models. (c-h) Sampled test trajectory and its total energy for the (c-d) ODEN, (e-f) HODEN, and (g-h) TRS-ODEN.

we evaluate the sample efficiency and find that the combination of two inductive bias improves the learning process more reliable (see Figure 4). For a more detailed analysis of reasoning on the improvement made by TRS-HODENs, see Section C in the supplementary material. Also, we find that TRS-ODENs and TRS-HODENs can predict critical points (stable centers and homoclinic orbits) of non-linear oscillators. See Section D in the supplementary material for details.

**Experiment III: Forced non-linear oscillator.** We set $\alpha = -0.2$, $\beta = 0.2$, $\gamma = 0$, and $\delta = 0.15$ for system parameters. Due to the periodic driving force $\delta \cos t$, this system is non-autonomous. Therefore, we use a tuple $(\mathbf{q}, \mathbf{p}, t)$ as an input of the neural networks for this experiment[10]. Hyperparameters of neural networks are same as them for Experiment II, except for $\lambda$: $\lambda \in \{0.5, 1, 5\}$ is estimated in here. We use the fourth order RK method for Solve, since it is not straightforward to apply the leapfrog solver for non-autonomous systems. We generate 200 and 50 trajectories whose lengths are 50 and 100, respectively, for train and test sets in this experiment, considering the complexity of the target system. Also, we reduce the noise to training dataset as $0.05n, n \sim \mathcal{N}(0, 1)$.

We find that TRS-ODENs clearly outperform their baselines with significant margin in both trajectory and energy MSE metrics (see Figure 5 (a-b)). From Figure 5 (c-h), one can check the dynamics predicted by the ODEN or HODEN diverge as times passes, while the TRS-ODEN shows reliable long-term behaviors. As a result, the total energy of the TRS-ODEN follows the ground truth reasonably, while that estimated by baselines soars explosively in $t > 6$.

**Experiment IV: Damped oscillator.** We simulate damped oscillators by setting the system parameters as follows: $\alpha = 1$, $\beta = 0$, $\gamma = 0.1$, $\delta = 0$. In this experiment, we assume the time-reversal symmetry tends to hold as $t \to \infty$, thus evaluate the time-dependent $\lambda$ approach. This assumption is

Table 1: Summary of test MSEs of Duffing oscillator experiments. All MSE values are multiplied by $10^2$.

| Metric | Model | Experiment I | Experiment II | Experiment III | Experiment IV |
|---|---|---|---|---|---|
| Traj. | ODEN | $4.05 \pm 2.66$ | $50.81 \pm 26.80$ | $39.21 \pm 21.19$ | $1.28 \pm 0.82$ |
| | HODEN | $0.84 \pm 0.37$ | $17.40 \pm 17.74$ | $24.09 \pm 14.29$ | $3.68 \pm 2.19$ |
| | TRS-ODEN | $\mathbf{0.31 \pm 0.19}$ | $13.78 \pm 14.86$ | $\mathbf{6.50 \pm 5.59}$ | $\mathbf{0.12 \pm 0.06}$ |
| | TRS-HODEN | N/A | $\mathbf{10.85 \pm 12.62}$ | N/A | N/A |
| Energy | ODEN | $9.04 \pm 10.14$ | $6.14 \pm 9.13$ | $446.61 \pm 304.91$ | $1.04 \pm 1.17$ |
| | HODEN | $0.08 \pm 0.09$ | $\mathbf{0.22 \pm 0.17}$ | $211.02 \pm 218.64$ | $8.26 \pm 9.60$ |
| | TRS-ODEN | $\mathbf{0.07 \pm 0.09}$ | $0.53 \pm 0.75$ | $\mathbf{5.94 \pm 4.00}$ | $\mathbf{0.03 \pm 0.03}$ |
| | TRS-HODEN | N/A | $0.29 \pm 0.18$ | N/A | N/A |

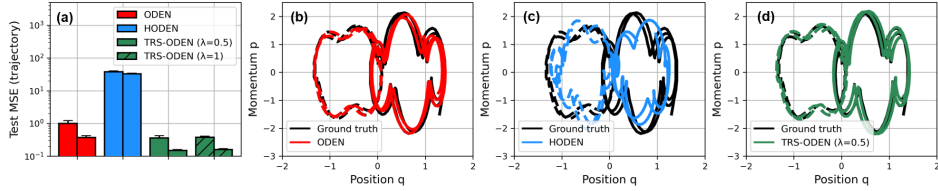

Figure 7: **Summary of Experiment V.** (a) Test trajectory MSEs across the models (left: mass 1 / right: mass 2). (b-d) Test trajectories from the (b) ODEN, (c) HODEN, and (d) TRS-ODEN (solid: mass 1 / dashed: mass2).

quite reasonable for various disspative irreversible systems, because their irreversibility is typically originated from the (odd powers of) $\mathbf{p}$[11] in their governing ODEs, e.g., $\gamma \mathbf{p}$ in (12). Since dissipative systems lose their kinetic energy as time passes, i.e., $\mathbf{p} \to 0$ as $t \to \infty$, we can design $\lambda$ as a linear increasing function of min-max normalized $t$. In this experiment, we evaluate four cases of $\lambda$: $\lambda \in \{0.5, 0.5t, 1, t\}$. Other hyper-parameters are same with them of Experiment I.

It is shown that the TRS-ODENs can outperform ODEN and HODEN, except for $\lambda = 1$ case (see Figure 6 (a-b)). Especially, $\lambda = 0.5t$ case shows great predictability in both time evolution and total energy of the damped system, while the ODEN loses its energy too excessively and the HODEN conserves its energy too strictly (see Figure 6 (c-h)). We believe it is owing to the balance between physics-based inductive bias and data-driven learning process. We summarize test MSEs of all Duffing oscillator experiments in Table 1.

## 4.2 Experiment V: Learning the real-world dynamics

We also conduct an experiment with real-world data from [37] to test whether models can learn the accurate dynamics for future behaviors even in real-world problems. This data consists of a measured trajectory of coupled double oscillators, which are neither conservative nor reversible due to the damping, coupling, measurement errors and other non-ideal effects. We use the first 3/5 of the trajectory for training, and the remains for test.

We use single hidden layer neural networks with 1,000 hidden units and `tanh` activations for all models. Figure 7 and Table 2 clearly show that the proposed TRS-ODEN outperforms baselines, especially the HODEN. It reveals 1) while enforcing the conservation may not be a good inductive bias for real world, 2) guiding time-reversal symmetry is helpful for model generalization.

Table 2: Summary of test MSEs of the real-world experiment (repeated 5 times). All MSE values are multiplied by $10^2$.

| Model | Mass 1 MSE | Mass 2 MSE |
|---|---|---|
| ODEN | $1.00 \pm 0.22$ | $0.37 \pm 0.05$ |
| HODEN | $38.13 \pm 2.16$ | $32.60 \pm 1.96$ |
| TRS-ODEN | $\mathbf{0.36 \pm 0.06}$ | $\mathbf{0.15 \pm 0.01}$ |

## 4.3 Experiment VI: Learning the chaotic strange attractors

Learning and predicting strange attractors are challenging tasks due to their chaotic behaviors. Time-reversal symmetry inductive bias can be helpful to model strange attractors, considering some of them are reversible when they consist of symmetric attractor/repellor pairs [41]. We evaluate our

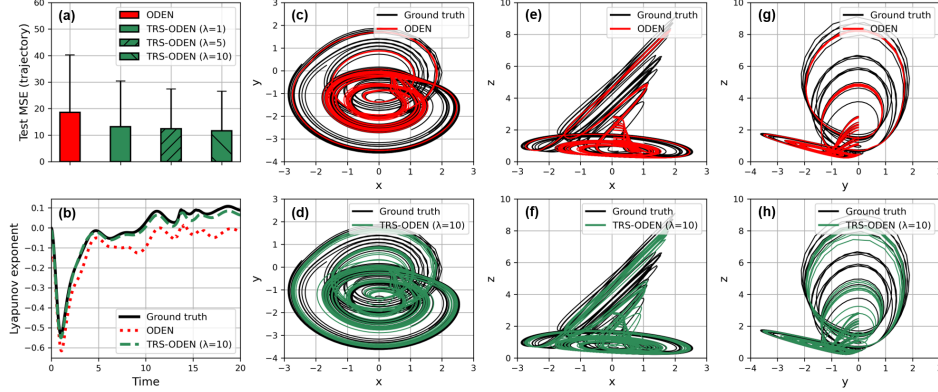

Figure 8: **Summary of Experiment VI.** (a) Test trajectory MSEs across the models. (b) Time evolutions of the Lyapunov exponents obtained from the ground truth, ODEN, and TRS-ODEN. (c-h) Sampled test trajectories in (c-d) $x$-$y$, (e-f) $x$-$z$, and (g-h) $y$-$z$ planes, predicted by the (c, e, g) ODEN and (d, f, h) TRS-ODEN.

framework for such a reversible strange attractor that is given by:

$$\frac{dx}{dt} = 1 + yz, \; \frac{dy}{dt} = -xz, \; \frac{dz}{dt} = y^2 + 2yz, \; x, y, z \in \mathbb{R}. \tag{13}$$

Note that this system shows reversal symmetry with non-trivial reversing operators $R : R(x, y, z) = R(-x, -y, -z)$. We generate 1,000 and 50 trajectories of (13) for training and test dataset, respectively, with sampling $z(t_0)$ randomly from uniform distribution $[1, 3]$ while fixing $x(t_0) = y(t_0) = 0$. Even with fixed $x(t_0)$ and $y(t_0)$, this task is still a interesting challenge considering the chaotic behaviors of strange attractors that are highly sensitive to the initial conditions [20]. We set the trajectory lengths of both training and test dataset to 400, with regular time-step size of 0.05. We add Gaussian noise $0.05n, n \sim \mathcal{N}(0, 1)$ to training trajectories.

We use two-hidden layer neural networks with 200 hidden units and `tanh` activations. Since it is not straightforward to set Hamilton's equation for (13), HODENs are not evaluated in here. We set a mini-batch size to 1,024. We use the fourth order RK method for `Solve`. After the evaluation, we find TRS-ODENs consistently outperform the ODEN (see Figure 8 (a) and Figure 8 (c-h) for quantitative and qualitative analysis, respectively). In addition, we calculate the Lyapunov exponent [44]:

$$\sigma_{\text{Lyapunov}} = \frac{1}{t_i - t_0} \log \frac{\|\delta \mathbf{x}(t_i)\|_2}{\|\delta \mathbf{x}(t_0)\|_2}, \tag{14}$$

where $\|\delta \mathbf{x}(t_i)\|_2$ is equal to the distance between two evolved states whose initial separation is infinitesimally small, i.e., $\|\delta \mathbf{x}(t_0)\|_2 \to 0$. From its definition, the Lyapunov exponent $\sigma_{\text{Lyapunov}}$ indicates a sensitivity on initial states for given dynamical system. Thus, it is used to detect and investigate the characteristics of chaotic systems: generally, larger positive $\sigma_{\text{Lyapunov}}$ means the system is more chaotic. Figure 8 (b) shows the time evolution of $\sigma_{\text{Lyapunov}}$ for test samples obtained from the TRS-ODEN matches well with that of ground truth, while the ODEN underestimates $\sigma_{\text{Lyapunov}}$.

## 5  Conclusion

Introducing physics-based inductive bias for neural networks is actively studied. e.g., ODE [6], Hamiltonian [15, 34, 43, 46, 7], and other domain knowledge [38, 39, 28, 31]. We have proposed a simple yet effective approach to incorporate the time-reversal symmetry into ODEN, coined TRS-ODEN, which is not shown in previous works. The proposed method can learn the dynamical system accurately and efficiently. We have validated our proposed framework with various experiments including non-conservative and irreversible systems.

There are some papers discuss the use of symmetry for neural networks. For example, the rotational or reflection symmetries are frequently used in computer vision tasks [13, 10, 45]. Some researchers have focused on finding symmetries using neural networks, especially in theoretical physics [9, 26, 3]. Among them, [3, 26] are closely related to our work because they discuss the method of searching a canonical transformation that satisfies the symplectic symmetry of Hamiltonian systems. Combining these approaches, i.e., finding symmetry, with our proposed framework, i.e., exploiting symmetry, would be an interesting direction for future work.

## Broader Impact

In this paper, we introduce a neural network model that regularized by a physics-originated inductive bias, the symmetry. Our proposed model can be used to identify and predict unknown dynamics of physical systems. In what follows, we summarize the expected broader impacts of our research from two perspectives.

**Use for current real world applications.** Predicting dynamics plays a important role in various practical applications, e.g., robotic manipulation [16], autonomous driving [25], and other trajectory planning tasks. For these tasks, the predictive models should be highly reliable to prevent human and material losses due to accidents. Our propose model have a potential to satisfy this high standard on reliability, considering its robustness and efficiency (see Figure 4 as an example).

**First step for fundamental inductive bias.** According to the CPT theorem in quantum field theory, the CPT symmetry, which means the invariance under the combined transformation of charge conjugate (C), parity transformation (P), and time reversal (T), exactly holds for all phenomena of physics [21]. Thus, the CPT symmetry is a fundamental rule of nature: that means, it is a fundamental inductive bias of deep learning models for natural science. However, this symmetry-based bias has been unnoticed previously. We study one of the fundamental symmetry, the time-reversal symmetry in classical mechanics, as a proof-of-concept in this paper. We expect our finding can encourage researchers to focus on the fundamental bias of nature and extend the research from classical to quantum, and from time-reversal symmetry to CPT symmetry. Our work would also contribute to bring together experts in physics and deep learning in order to stimulate interaction and to begin exploring how deep learning can shed light on physics.

## Acknowledgments and Disclosure of Funding

The authors received no third party funding for this work.

## Footnotes

[1]Code is available at https://github.com/inhuh/trs-oden.

[2]For Hamiltonian as an example, $\mathbf{x} = (\mathbf{q}, \mathbf{p})$, where $\mathbf{q} \in \mathbb{R}^n$ and $\mathbf{p} \in \mathbb{R}^n$ are positions and momenta.

[3]Note that the most basic definition of the Hamiltonian is the sum of kinetic and potential energy, i.e., $\mathcal{H}(\mathbf{q}, \mathbf{p}) = \mathbf{p}^2/2 + V(\mathbf{q})$ (if we omit the mass) [14], which possess $\mathcal{H}(\mathbf{q}, \mathbf{p}) = \mathcal{H}(\mathbf{q}, -\mathbf{p})$ naturally.

[4]Let's consider a damped pendulum. They are irreversible since one can distinguish the motion of the pendulum in forward (amplitude increases) and that in backward directions (amplitude decreases).

[5]TRS-ODENs require approximately $2\times$ larger training time than the vanilla/default ODENs because the backward as well as forward evolutions need to be calculated.

[6]They can be calculated from trajectories. For example, a total energy of simple oscillator is $\mathbf{q}^2 + \mathbf{p}^2$.

[7]Typically, Duffing oscillator is given by a second order ODE $\ddot{\mathbf{x}} + \alpha\mathbf{x} + \beta\mathbf{x}^3 + \gamma\dot{\mathbf{x}} = \delta\cos(t)$. We separate this equation from the perspective of the pseudo-phase space, although they are not in canonical coordinates.

[8]One can force the TRS-ODEN to be more symmetric by increasing the regularization strength $\lambda$. See Section B in the supplementary material for details.

[9]It can be obtained straightforwardly by combining (5) and (8), similar to (9-10).

[10]In [15, 7], the authors say for HODENs, time dependency should be modeled separately from them. However, we use time-dependent HODENs in here to prevent large modifications of HODENs for fair comparison.

[11]It is because of the definition of the classical reversing operator $R(\mathbf{q}, \mathbf{p}) = (\mathbf{q}, -\mathbf{p})$.

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
