[Supplementary Material]

# Supplementary Material:
# Time-Reversal Symmetric ODE Network

## A  Time-reversal symmetry loss for non-autonomous systems

Here, we consider the time-reversal symmetry of non-autonomous ODE systems, i.e., systems that depend on time $t$ explicitly as follows:

$$\frac{d\mathbf{x}}{dt} = f(\mathbf{x}, t). \tag{S1}$$

This non-autonomous systems are said to be time-reversal symmetric if there is a reversing operator $R_a : (\mathbf{x}, t) \mapsto (R(\mathbf{x}), -t + a)$ which satisfies [23]:

$$\frac{dR(\mathbf{x})}{dt} = -f(R(\mathbf{x}), -t + a), \tag{S2}$$

for some $a \in \mathbb{R}$. It means that we should consider the time $t$ itself carefully, as well as the direction of time, unlike the autonomous case (6-10) in the main paper. For example, consider forced non-linear oscillators estimated in Experiment III in the main paper:

$$\frac{d\mathbf{q}}{dt} = \mathbf{p}, \ \frac{d\mathbf{p}}{dt} = -\alpha\mathbf{q} - \beta\mathbf{q}^3 + \delta\cos(\omega t + \phi). \tag{S3}$$

(S3) is time-reversal symmetric under $R_{-2\phi/\omega} : (\mathbf{q}, \mathbf{p}, t) \mapsto (\mathbf{q}, -\mathbf{p}, -t - 2\phi/\omega)$.

The forward time evolution of non-autonomous ODENs is given by:

$$\tilde{\mathbf{x}}(t_{i+1}) = \mathtt{Solve}\{\tilde{\mathbf{x}}(t_i), t_i, f_\theta, \Delta t_i\}, \ \tilde{\mathbf{x}}(t_0) = \mathbf{x}(t_0). \tag{S4}$$

On the other hand, the backward time evolution is equal to:

$$\tilde{\mathbf{x}}_R(\tau_{i+1}) = \mathtt{Solve}\{\tilde{\mathbf{x}}_R(\tau_i), \tau_i, f_\theta, -\Delta t_i\}, \ \tilde{\mathbf{x}}_R(\tau_0) = R_a(\tilde{\mathbf{x}}(t_0)), \tag{S5}$$

where $\tau_i = -t_i + a$. As a result, the time-reversal symmetry loss of autonomous ODE systems is given by:

$$\mathcal{L}_{\text{TRS}} \equiv \sum_{i=0}^{T-1} \|R(\mathtt{Solve}\{\tilde{\mathbf{x}}(t_i), t_i, f_\theta, \Delta t_i\}) - \mathtt{Solve}\{\tilde{\mathbf{x}}_R(\tau_i), \tau_i, f_\theta, \tau_i\}, -\Delta t_i\}\|_2^2. \tag{S6}$$

## B  Guaranteeing time-reversal symmetry by increasing $\lambda$

Minimizing $\mathcal{L}_{\text{TRS-ODEN}}$ ((11) in the main paper) guides, but does not guarantee the perfect time-reversal symmetric solution. Nevertheless, one can force the solution be almost symmetric by increasing the regularization strength $\lambda$. To confirm this, we evaluate the relative error between the forward and backward time evolutions of TRS-ODENs, that trained with varying $\lambda$. Experimental setting is equal to that used in Experiment I in the main paper. It shows that large $\lambda = 10^3$ guarantees lower than $10^{-3}$ relative error, i.e., almost perfect time-reversal symmetry, without any performance degradation, as shown in Figure S1.

Figure S1: $\lambda$ vs. relative error between forward and backward time evolutions.

Figure S2: (a) Calculated $\mathcal{H}(\mathbf{q}, \mathbf{p}) - \mathcal{H}(\mathbf{q}, -\mathbf{p})$ of ground truth, HODEN, and TRS-HODEN. (b) Comparison of kinetic energy profiles obtained from ground truth, HODEN, and TRS-HODEN. Note that we calibrate the ground energy level to make $\mathcal{H}(\mathbf{0}, \mathbf{0}) = 0$ for all models.

Figure S3: Hamiltonian surfaces obtained from the (a) ground truth, (b) HODEN, and (c) TRS-HODEN. The ground truth Hamiltonian shows symmetric double well shape.

## C   Reasoning on the improvement made by TRS-HODENs

As mentioned in Section 3.1 in the main paper, the Hamiltonian $\mathcal{H}$ of conservative and reversible systems satisfies $\mathcal{H}(\mathbf{q}, \mathbf{p}) = \mathcal{H}(\mathbf{q}, -\mathbf{p})$. With this symmetry property, we analyze the reason of improvement made by TRS-HODENs over HODENs in Experiment II in the main paper. Note that the ground truth Hamiltonian of non-linear oscillator tested in Experiment II is described as:

$$\mathcal{H}(\mathbf{q}, \mathbf{p}) = \frac{\mathbf{p}^2}{2} + \frac{\alpha \mathbf{q}^2}{2} + \frac{\beta \mathbf{q}^4}{4}, \tag{S7}$$

which clearly possesses $\mathcal{H}(\mathbf{q}, \mathbf{p}) = \mathcal{H}(\mathbf{q}, -\mathbf{p})$.

We find that the time-reversal symmetry loss helps the learned $\theta$-parameterized Hamiltonian $\mathcal{H}_\theta(\mathbf{q}, \mathbf{p})$ possess the above property thanks to the symmetry under the momentum-reversing operator $R(\mathbf{q}, \mathbf{p}) = (\mathbf{q}, -\mathbf{p})$. To show this, we calculate $\mathcal{H}_\theta(\mathbf{q}, \mathbf{p}) - \mathcal{H}_\theta(\mathbf{q}, -\mathbf{p})$ for the HODEN and TRS-HODEN ($\lambda = 10$) tested in Experiment II, with varying $\mathbf{p}$ from 0 to 1.5 and fixing $\mathbf{q}$ to 0 (see Figure S2 (a)). It shows that the Hamiltonian of the HODEN does not follow $\mathcal{H}(\mathbf{q}, \mathbf{p}) = \mathcal{H}(\mathbf{q}, -\mathbf{p})$ precisely, while that of the TRS-HODEN is almost even function of $\mathbf{p}$. As a result, TRS-HODENs can learn the ground truth Hamiltonian from noisy data more structurally and efficiently.

To confirm the above discussion, we compare their kinetic energies $K_{\theta_1}(\mathbf{p}) = \mathcal{H}_\theta(\mathbf{0}, \mathbf{p}) + \text{const.}$ (see Figure S2 (b)). It shows the kinetic energy of the TRS-HODEN is almost indistinguishable from that of the ground truth. On the other hand, the kinetic energy of the HODEN does not match well with that of the ground truth. In Figure S3, We plot the Hamiltonian (total energy) surfaces across the models. One can check the Hamiltonian surface of the HODEN shows highly asymmetric double well shape, unlike that of the ground truth and TRS-HODEN.

Figure S4: The critical phase space trajectories obtained from the (a) ODEN, (b) HODEN, (c) TRS-ODEN, and (d) TRS-HODEN.

Table S1: Summary of phase space trajectory and total energy MSEs evaluated in Section D. All MSE values are multiplied by $10^2$.

| Model | ODEN | HODEN | TRS-ODEN | TRS-HODEN |
|---|---|---|---|---|
| MSE (Traj.) | $14.28 \pm 10.47$ | $15.26 \pm 25.15$ | $3.88 \pm 5.92$ | $\mathbf{2.03 \pm 2.17}$ |
| MSE (Energy) | $9.31 \pm 16.11$ | $0.32 \pm 0.53$ | $0.52 \pm 0.78$ | $\mathbf{0.21 \pm 0.21}$ |

## D Predicting stable centers and homoclinic orbits of non-linear oscillators

The non-linear oscillator systems estimated in Experiment II in the main paper have two stable centers at $(1, 0)$, $(-1, 0)$, and saddle point at $(0, 0)$ (see Figure S3 (a)). Clearly, at the centers, states do not evolve with time at all, i.e., the stable equilibrium states. Near the saddle point, there are two interesting trajectories, that appear to start and end at the same saddle point. These trajectories are called homoclinic orbits [42]. Note that the homoclinic orbits lie on $\mathbf{q} > 0$ and $\mathbf{q} < 0$ respectively start from $(\epsilon, \epsilon)$ and $(-\epsilon, -\epsilon)$, for some small positive constants $\epsilon$.

Here, we estimate whether the learned dynamics can represent the special trajectories originated from these critical points well. To do this, we generate trajectories, whose initial states are given by the centers or saddle point[1], by using the models trained in Experiment II: ODEN, HODEN, TRS-ODEN ($\lambda = 10$), and TRS-HODEN ($\lambda = 10$). Figure S4 demonstrates the generated phase space trajectories. For the ODEN, it cannot achieve the accurate time evolution at all. HODEN shows relatively reasonable behaviors, but they predict the same direction of homoclinic orbits for $(\epsilon, \epsilon)$ and $(-\epsilon, -\epsilon)$. Also, periodic motions near the centers are observed for HODENs. TRS-ODEN and TRS-HODEN show two separated homoclinic orbits clearly. Moreover, the TRS-HODEN shows stable equilibrium behaviors at the center points. In summary, the TRS-HODEN can predict physically-consistent behaviors even for critical points. We summarize the phase space trajectory and total energy MSE metrics in Table S1.

## Footnotes

[1] We use $10^{-8}$ and $10^{-2}$ instead of 0 and $\epsilon$, respectively, considering numerical stability.