[Reviews · NeurIPS 2020]

Review 1

Summary and Contributions: In this paper, the authors propose a novel NN model that incorporate the time-reversal symmetry into ODE networks. They develop a loss for this model. These can be combined with variants of ODE networks, such as Hamiltonian ODE networks. They empirically validate their model in several setups, i.e., conservative vs. non-conservative and reversible vs. irreversible systems.

Strengths: The time-reversal symmetry is the important symmetry in physical systems. And, I think the proposed scheme is a reasonable and principled way to incorporate this important property into recently-attention-getting ODE networks. Empirical results show their model work reasonably for simple physical systems.

Weaknesses: I think (not major) weak points of this paper is twofold: The first one is in the empirical evaluations. They use one data generating system and deal with setups (eg conservative vs. non-conservative) by varying parameters with Gaussian noises. Although I agree the proposed model works to learn the system, it is difficult to see the empirical properties, limitations and so on from this experiment. And the second is the part in their loss where the balance between two type of losses is controlled by the regularization term. To be honest, I am not dissatisfied with this part and wonder what is the premise of this part.

Correctness: Because the basic idea is simple (principled), I found no concern in the correctness about their descriptions in the paper.

Clarity: The paper is well-written with sufficient information, and the readability is high.

Relation to Prior Work: The paper describes clearly the relation with existing works and their contributions upon those.

Reproducibility: Yes

Additional Feedback:


Review 2

Summary and Contributions: The authors present a regularizer for learning approximately time-symmetric governing equations for ordinary dynamical systems. They demonstrate that using the proposed regularization results in improved learning of dynamics from noisy data. The paper's main contribution is the recognition that time-symmetry may be an important inductive bias for learning real-world physical systems.

Strengths: The work is well presented. It is novel and of interest to anyone learning dynamics from observed data.

Weaknesses: The principle limitation of this work is that, like so many things in machine learning, we don't have real guarentees. The regularization is presented as a strength, and it is when we would like something that is "near" a time-symmetric solution. But a solution that guarentees it would be very welcome. Other smaller, more specific points: How general is this choice of R? How difficult is it to choose the correct R? What are the consequences of choosing the wrong R? What is the impact on training time or convergence?

Correctness: The approach and evaluation appear to be correct.

Clarity: The paper is very well written and easy to follow. The authors should explicitly state the definition of R, presumable (p,-q), they used in their experiments. The authors clearly state that their data generation model is time-reversible whenever gamma=delta=0. A small type: line 190 "lean" -> "learn"?

Relation to Prior Work: The authors try to delineate their work from previous works. However, they appear to be setting themselves up to differentiate their work from approaches that use regularizations to encode physical knowledge (lines 27,28). Yet, the authors' work comes down to exactly a regularization that encodes physical knowledge. On line 29 the authors suggest that these approaches are somehow problem specific and thus don't generalize, the implication being that their method does. The authors should be more explicit as to what these other regularizers are and how specific they might be.

Reproducibility: Yes

Additional Feedback: I found the rebuttal to be solid. The authors have noted several additional experiments and points developed during the rebuttal period. They have addressed my criticisms and as a result, I am more firm in my score.


Review 3

Summary and Contributions: This paper proposes a new method for learning ordinary differential equations that underlie observed time series that biases the ordinary differential equations towards being time-reversal symmetric.

Strengths: The paper was well-written and easy to follow.

Weaknesses: With the new experiments, the main weaknesses that I previously had here have been answered. I think the only remaining weakness is that this will be most applicable to classical mechanics, but those systems are already often well-understood much of the time from basic physics, and so one could instead learn a dimensionality-reduced model of the enlarged physics model rather than using this method.

Correctness: Yes, yes.

Clarity: Yes.

Relation to Prior Work: Yes.

Reproducibility: Yes

Additional Feedback:


Review 4

Summary and Contributions: This paper proposes a Time-reversal symmetric ODE network that utilizes a novel loss function measuring how well the ODE network complies with time-reversal symmetry. The main idea is to use the time-reversal symmetry of classical dynamics to measure the discrepancy in the time evolution of ODE networks between the forward and backward dynamics. The proposed time-reversal symmetric ODE is shown to be more sample efficient and achieves better prediction accuracy comparing with vanilla ODE neural network, Hamiltonian ODE neural networks. ------------------------------------------------------------------------------------------------------ I think the rebuttal addressed my main concern which is lack of evaluation of real-world datasets. I'd like to keep my score (7-accept).

Strengths: 1) The paper provides a simple but very effective way to incorporating physics-based bias for neural networks. 2) The paper provides very thorough empirical evaluation of the proposed neural network on data generated from physics systems with different characteristics

Weaknesses: 1) The paper did not provide any evaluation results of the system on real-world data

Correctness: The claims and methods seems to be correct

Clarity: The paper is generally well-written with some minor grammatical errors. Suggestions: 1) line 13: "better predictive errors" -> "smaller predictive errors" or "better predictive performance" 2) line 106: "Furthermore, they can lean" -> "Furthermore, they can learn" 3) line 107: "because they fully exploit the nature of" -> "since they fully exploit the nature of"

Relation to Prior Work: The paper clearly discussed how this work differs from previous contributions, such as Hamiltonian ODE neural network, which do not work properly for non-conservative systems.

Reproducibility: Yes

Additional Feedback:

[Author Response · NeurIPS 2020]



**Figure A:** Test trajectory of oscillators (solid: mass 1 / dashed: mass 2) predicted by (a) ODEN, (b) HODEN, and (c) TRS-ODEN.

We thank all reviewers for their valuable comments. In particular, we appreciate that reviewers agree on the importance
of the time-reversal symmetry (TRS) [R1, R2, R4] and clear writing [R1, R2, R3, R4]. Followings are our responses.

**Q1. [R1, R3, R4] Empirical evaluation on real dataset.** During the
rebuttal period, we performed an experiment with real-world data [35].
This data consists of a trajectory of real double oscillators, which are
neither conservative nor reversible due to damping and other non-ideal
effects. We set the first 3/5 of the trajectory for training, and the remains
for test. We used same hyper-parameters in Experiment IV. **Table A**

**Table A:** Experimental results (repeated 5 times).

| Model | Mass 1 MSE | Mass 2 MSE |
|---|---|---|
| ODEN | $1.00 \pm 0.22$ | $0.37 \pm 0.05$ |
| HODEN | $38.13 \pm 2.16$ | $32.60 \pm 1.96$ |
| **TRS-ODEN** | $\mathbf{0.36 \pm 0.06}$ | $\mathbf{0.15 \pm 0.01}$ |

and **Figure A** show our model outperforms baselines, especially HODEN. It reveals 1) while enforcing the conservation
may not be good for real world, 2) guiding symmetry with TRS loss is helpful. We will add this result in the final draft.

**Q2. [R1] Balance between two losses.** Because the total loss is given by $\mathcal{L}_{\text{ODE}} + \lambda \cdot \mathcal{L}_{\text{TRS}}$, higher (lower) $\lambda$ leads
stricter (looser) symmetry. While $\lambda$ is treated as a constant generally, one can deal with it as a function of $(\mathbf{q}, \mathbf{p})$ and $t$,
with the assumption that the irreversible term in ODE is also a function of them. We premise it gives more precise
balance between two losses, especially when addressing irreversible systems. We will clarify this part in the final draft.

**Q3. [R2] Guarantees on TRS.** We agree with the comment that regularizers do
not guarantee the perfect TRS solution. Nevertheless, one can force the solution
be almost symmetric by increasing $\lambda$. To confirm this, we evaluated the relative
error between forward and backward trajectories of TRS-ODENs that trained
with varying $\lambda$ (**Figure B**). It shows large $\lambda = 10^3$ guarantees lower than $10^{-3}$
relative error, without performance degradation. We believe the flexibility to
control the degree of TRS is rather a strong merit of TRS-ODEN, as mentioned in
line 56-61 and 129-131. We will add the respective discussion to the final draft.

**Figure B:** $\lambda$ *vs.* test relative error/MSE.

**Q4. [R2] The reversing operator.** We used the reversing operator $R : R(\mathbf{q}, \mathbf{p}) = (\mathbf{q}, -\mathbf{p})$, which we will state
explicitly in the final draft. It is quiet general for classical mechanics, whose $\mathbf{p}$ is naturally negated by the time-reversal,
as described in line 141-145. However, there are some particular systems that such $R$ does not work well, e.g., chaotic
strange attractors (see **Q8** for more information). While one can set proper $R$ for this case (as we did in **Q8**), automatic
search of $R$ from unknown data is an interesting future work. We will add a related discussion in the final draft.

**Q5. [R2] Impact on training time.** TRS-ODENs require approximately two times larger training time than default
ODENs because the backward as well as forward evolutions need to be calculated. We will state it in the final draft.

**Q6. [R2] Previous works.** The regularizers mentioned in line 27-29 use very specific models such as Navier-Stokes
equation, thus can be applied only when governing laws are exactly known. We will clarify this part in the final draft.

**Q7. [R3] Motivation and benefit of TRS loss.** As described in line 39-47 and pointed by R1, R2, and R4, TRS is
an important symmetry in physics. It motivates us to design the TRS regularizer. As you pointed out, it is important
to inform readers what kinds of systems that TRS works well. To do this, we stated the target systems and expected
benefits of TRS for them in line 111-131. Clearly, TRS is powerful to model reversible systems, which are natural in
classical dynamics. Thus, they are the primary target of our proposed method. In addition, we would like to emphasize
that even for irreversible systems TRS is helpful for model generalization, as shown in Experiment IV and real-world
experiment (**Q1**, **Table A** and **Figure A**). We analyze the reason as twofold: 1) the irreversible terms in ODE can be
(partially) negligible during dynamics, and 2) TRS regularizer flexibly guides the symmetry rather than enforcing it.

**Q8. [R3] Strange attractors.** Some strange attractors show TRS
under non-trivial reversing operators $R$, according to "Sprott, J. C.
*International Journal of Bifurcation and Chaos*, 25(05):1550078,
2015". During the rebuttal period, we conducted an experiment
with $\dot{x} = 1 + yz, \dot{y} = -xz, \dot{z} = y^2 + 2yz$, a reversible strange
attractor under $R : R(x, y, z) = (-x, -y, -z)$. Since it is not
straightforward to set Hamilton's equation for this system, HODEN
is not evaluated. We generated 1,000 and 50 trajectories for training
and test, respectively. As a result, we found TRS-ODEN can

**Figure C:** Sampled test trajectories of the strange attractors predicted by (a) ODEN and (b) TRS-ODEN.

achieve smaller test MSE than ODENs, e.g., from 17.0 to 13.2 (see **Figure C**). We will add this result to the final draft.

**Q9. [R2, R4] Suggestions on clarity.** We will revise typos and grammatical errors according to reviewers' suggestions.

[Meta-Review · NeurIPS 2020]

The reviewers agree that this paper represent a strong contribution to the community. The reviewers concerns related to lack of evaluation were addressed by the authors in the rebuttal.